# *Bacillus simplex* as the Most Probable Culprit of Penetrating Trauma Infection: A Case Report

**DOI:** 10.3390/pathogens11101203

**Published:** 2022-10-18

**Authors:** Panagiota Xaplanteri, Dimitrios S. Serpanos, Ellie Dorva, Tatiana Beqo-Rokaj, Eleni Papadogeorgaki, Alexandra Lekkou

**Affiliations:** 1Department of Microbiology, General Hospital of Eastern Achaia, 25001 Kalavrita, Greece; 2Department of Internal Medicine, General Hospital of Eastern Achaia, 25001 Kalavrita, Greece; 3Department of Primary Healthcare, General Hospital of Eastern Achaia, 25001 Kalavrita, Greece; 4Central Diagnostic Laboratories, Hygeia Hospital Athens, 15123 Athens, Greece; 5Department of Internal Medicine, University General Hospital of Patras, 26504 Patras, Greece; 6Department of Infectious Diseases, University General Hospital of Patras, 26504 Patras, Greece

**Keywords:** *Bacillus simplex*, penetrating trauma, infection

## Abstract

*Bacillus simplex* is an environmental microorganism found in soil. Herein, we present the case of a 69-year-old Greek male patient who attended the Emergency Department at our hospital. The patient complained of lower right extremity swelling and pain, after suffering penetrating trauma whilst doing farm work in a rural area. Swab aerobic cultures revealed *Bacillus simplex* as identified by MALDI-TOF Mass Spectrometry. The strain was susceptible to vancomycin, imipenem, clindamycin, and ciprofloxacin. Our patient refused hospitalization; therefore, both ciprofloxacin and clindamycin were registered for a total of 19 days. No complications were experienced, and he recovered fully. In our case, the thorough cleaning of the ulcer bed prior to sample collection, the fact that it was the only microorganism isolated, and the wound’s aggravating mechanism led the authors of the present study to the conclusion that *B. simplex* was the most probable culprit of the infection. To our knowledge, this is the second probable case of *B. simplex* infection described worldwide, and the first in Greece.

## 1. Introduction

*Bacillus simplex* is an environmental microorganism found in soil [1]. Herewith presented is the case of a 69-year-old Greek male patient who attended the Emergency Department. The patient complained of lower right extremity swelling and pain, after suffering penetrating trauma whilst doing farm work in a rural area. Swab aerobic cultures revealed a microorganism with the shape and characteristics of *Bacillus*, identified by MALDI-TOF Mass Spectrometry as *Bacillus simplex*. To our knowledge this is the second probable case of *B. simplex* infection described worldwide, and the first in Greece.

## 2. Presentation of Case

A 69-year-old male Greek patient with a significant history of chronic obstructive pulmonary disease, hypertension and being an active smoker attended the Emergency Department of our hospital. He complained of lower right extremity swelling and pain, after suffering penetrating trauma whilst doing farm work in a rural area. Physical examination revealed cellulitis, a 10 cm diameter ulcer just above the ankle, tenderness, swelling, erythema, and intense edema from the middle of the leg to the tip of the foot. The patient was afebrile, blood pressure was 160/94 mmHg, pulse rate 99/min, Spo2 98%. The area of the ulcer was cleaned thoroughly, and swab cultures were obtained from the ulcer’s crater according to the Levine technique [2]. 

Due to the extended edema, Vascular Ultrasound was performed and proved negative for deep vein thrombosis. Blood cultures were not obtained. The patient did not accept hospital admission and was released with ciprofloxacin 500 mg twice a day and clindamycin 600 mg thrice a day whilst awaiting the swab cultures results. 

Laboratory findings revealed an elevated white blood cell count of 14300/mm^3^ (normal values 4000–10000/mm^3^), elevated C-reactive protein levels of 4.6 mg/dL (normal values 0.0–0.9 mg/dL), whereas serum glucose, renal and liver parameters were within normal range. 

The swab samples were inoculated on partitioned agar plates containing 5% sheep blood agar and McConkey agar number 3, by Thermo Scientific™ (Oxoid Ltd., Wade Road, Hampshire, UK). No direct Gram or Ziehl-Neelsen staining were performed from the ulcer. Swab cultures for anaerobes and fungus proved negative. Aerobic culture was positive onto blood agar plate after incubation at 37 °C overnight, with no swarming or hemolysis present (Figure 1a). Gram stain revealed a microorganism with the shape of Bacillus, catalase-positive, with no obvious endospores—if present, they did not alter the shape of the bacterium (Figure 1b). Gram stain was variable. MALDI-TOF Mass Spectrometry (Vitek MS BioMerieux, Athens, Greece, BioMerieux Hellas) identified the strain as *Bacillus simplex* with confidence value 99.9. The respective image of the bacterium identification is shown in Figure 2.

The breakpoint criteria for our strain were compared to those of *Staphylococcus* spp. according to the European Committee on Antimicrobial Susceptibility Testing (EUCAST) guidelines [3]. For disk diffusion susceptibility testing, an imipenem disk containing 10 mcg, clindamycin disk containing 2 mcg and ciprofloxacin disk containing 5 mcg were provided from Vaktro Scientific, Panepistimiou 195 Street, 26443, Patras, Achaia, Greece. Using the disk diffusion susceptibility testing our strain proved susceptible to imipenem, clindamycin and ciprofloxacin. As our strain proved susceptible to ciprofloxacin and clindamycin, the initial antibiotic scheme did not change. On follow up after 14 days of the antimicrobial agents’ registration, clinically there was significant improvement but there were still signs of infection. Therefore, the patient continued the therapy for a total of 19 days. No complications were experienced, and the patient recovered fully.

## 3. Discussion

*Bacillus* species are spore-forming rods that stain gram positive or gram variable and grow in aerobic conditions. Their natural habitat is the soil [4,5]. *Non-Bacillus anthracis* infections, most commonly due to *Bacillus cereus*, are related to food poisoning, ocular infections, deep-seated soft tissue infections, and systemic infections [5]. The taxonomic relationship between many *Bacillus* species is difficult to decipher as there is similarity in chromosomal structure and phenotypic characteristics among them [4,6]. The formation of endospores is the reaction of the *Bacillus* to inhospitable environmental conditions, such as extreme temperatures, radiation, and disinfectants [4].

Meyer and Gottheil first described *Bacillus simplex* in 1901 as an environmental microorganism found in soil [1]. *B. simplex* bacterial cells produce oval endospores [7]. In our sample no obvious endospores were detected with gram stain (Figure 1b). *B. simplex* presents a variable gram reaction. Most strains are strictly aerobic [7]. This applies also to our strain, which appeared both gram positive and gram negative and developed under aerobic conditions. 

Various virulence factors of *B. simplex* have been described. The microorganism produces a heat-stable toxin, like cereulide of *B. cereus*, that causes emesis in humans. This toxin inhibits mitochondrial activity and causes liver failure and cellular damage. It also inhibits human natural killer cells of the immune system [8]. *B. simplex* strain N65.1 also produces hemolysin BL (HBL), non-hemolytic enterotoxin NHE and CytK (or hemolysin IV) [9,10,11]. Hemolysins lyse erythrocytes. As a result, nutrients and iron-binding proteins are released for the pathogenic bacteria to use for their survival, and they provoke damage to epithelial cells, enhancing the microorganisms’ advance and access to deeper layers [11]. 

NHE has hemolytic activity towards erythrocytes from several mammalian species [11]. Both HBL and NHE are related to the hemolytic and cytotoxic toxin cytolysin A (ClyA, or HlyE or SheA), expressed during anaerobic growth of *Escherichia coli*, *Shigella flexneri* and *Salmonella enterica* serovars Typhi and Paratyphi A [8]. HBL component B and cytolysin A share a similar tertiary structure [8]. HBL has dermonecrotic activity and is cytotoxic towards Vero cells and retinal tissue [8]. NHE forms transmembrane pores which aggravate colloid osmotic lysis and cell death [8]. The pore-forming ability in phospholipid membranes of NHE is innate and consequently independent of a binding receptor [8]. CytK has cytotoxic, necrotic and hemolytic activities. It was first described in 2000 [11,12]. In *B. cereus* and *Bacillus cytotoxicus* strains, CytK relates to bloody diarrhea and increased mortality and has an amino-acid sequence 30% identical to α-hemolysin (α-toxin, Hla) of *Staphylococcus aureus* [11]. A-hemolysin of *S. aureus* is a well-described cause of dermonecrosis and lethal infection [13]. *S. aureus* α-toxin binds on the host cell membrane via its receptor ADAM10, and forms pores through the eukaryotic lipid bilayer. As a result, molecules with low molecular weight, like calcium ions, potassium ions and ATP leak through the pore, leading to cell lysis. The increased intracellular calcium triggers hydrolysis of membrane phospholipids and the conversion of arachidonic acid to leukotrienes, prostanoids, and thromboxane A2. In addition, there is alteration in the cell signaling pathways via the cytoplasmic tail of ADAM10, which induces NF-κB nuclear translocation and the production of proinflammatory cytokines IL-1β, and IL-6. The cascade of alterations that follow includes recruitment of immune cells, increased vascular permeability, tissue edema and inflammation. Pneumocytes and keratinocytes are key targets as well as epithelial and endothelial cells, T cells, monocytes, macrophages and neutrophils [13,14,15]. In addition, α-toxin hastens cell death of Th1 cells via Ca2+-mediated activation. This cell death is not induced via caspases’ activation [16]. *B. simplex* strain N58.2 has high cytotoxic potential, similar to *B. cereus* group III strains that have high cytotoxic activity [10]. Although there is little evidence in literature regarding the virulence of *B. simplex* strains, and most of the aforementioned toxins are best described as emesis-provoking toxins and food poisoning, their cytotoxic, dermonecrotic and inflammatory properties cannot be overlooked regarding soft tissue infection and inflammation and should be taken under consideration in our case.

To our knowledge, there is no reference in the literature regarding *B. simplex* susceptibility. However, there are studies regarding *B. cereus* strains. In the referred-to studies, antibiotic susceptibility was tested for ampicillin, cefazolin, cefotaxime, ceftazidime, imipenem, vancomycin, amikacin, gentamicin, erythromycin, levofloxacin, clindamycin, chloramphenicol and rifampin using the Clinical and Laboratory Standards Institute (CLSI) breakpoints as a guide [17,18]. In such studies, susceptibility was detected to imipenem, vancomycin, chloramphenicol, gentamicin, amikacin, linezolid, rifampin, levofloxacin and ciprofloxacin [5,17,18]. As expected, most strains were resistant to β-lactam antibiotics as there is genetical resistance of *B. cereus* to all beta lactams except carbapenems [18]. Regarding carbapenems, there are in vitro studies demonstrating that *B. cereus* strains possess metallo-beta lactamase genetically, therefore further elucidation is needed for the empirical use of these antibiotics [18]. Susceptibility to fluoroquinolones and clindamycin is strain-dependent and therefore should always be tested prior to administration [18]. Regarding daptomycin, there are in vitro studies showing that daptomycin resistance occurs under swarming condition [18]. Non-*B. cereus* strains are usually susceptible to imipenem, vancomycin, clindamycin and ciprofloxacin [17,19]. Erythromycin-resistant genes *ermD* and *ermK* localized on an 11.4-kbp plasmid have been reported in all *Bacillus licheniformis* strains resistant to erythromycin [19]. Resistance to erythromycin has also been reported for *Bacillus subtilis*, *Bacillus clausii* and *Bacillus anthracis* via methylation of the 23S rRNA macrolide binding sites [19]. *Bacillus subtilis* and *Bacillus sonorensis* strains have been reported susceptible to erythromycin [19]. *Bacillus subtilis* strains have been reported resistant to chloramphenicol, tetracycline, rifampin and streptomycin [19]. Disk diffusion susceptibility testing provides highly accurate results in *Bacillus* strains in comparison to microdilution [17]. There are studies which have revealed that the breakpoint criteria for susceptibility of Bacillus species were based on those respective for *Staphylococcus* spp. [17,18]. Therefore, the breakpoint criteria for our strain were compared to those of *Staphylococcus* spp. according to the European Committee on Antimicrobial Susceptibility Testing (EUCAST) guidelines [3]. Our strain, using the disk diffusion susceptibility testing, proved susceptible to imipenem, clindamycin and ciprofloxacin. As our strain proved susceptible to ciprofloxacin and clindamycin, the initial antibiotic scheme did not change. The patient had immediate clinical improvement and an uncomplicated course till full recovery. 

Isolation of *Bacillus* organism requires careful clinical evaluation to rule out possible contaminants [5]. *Bacillus* species are described as important pathogens when open wounds with contamination from the ground takes place [20]. 

To our knowledge this is the second case of *B. simplex* infection described worldwide and the first in Greece. The initial (Italian) case regards a brain abscess [7]. In another study, *B. simplex* strain P558 has been isolated from a fecal sample of a 25-year-old Saudi male as part of gut flora [21]. 

## 4. Conclusions

In our case of the male Greek patient, the thorough cleaning of the ulcer bed prior to sample collection, the fact that it was the only microorganism isolated, and the wound’s aggravating mechanism led the authors of the present study to the conclusion that *B. simplex* was the most probable culprit of the infection. 

## Figures and Tables

**Figure 1 pathogens-11-01203-f001:**
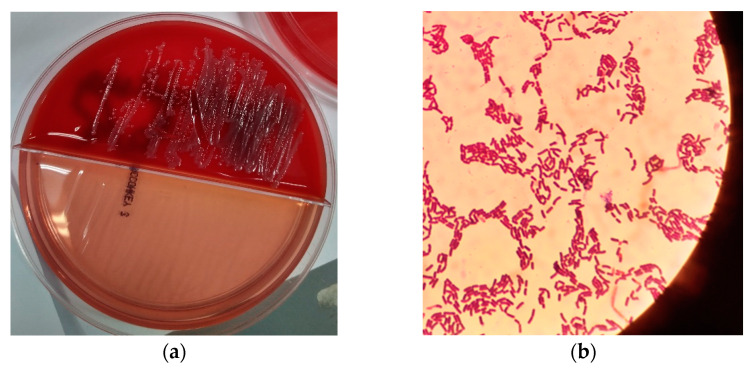
(**a**) Aerobic swab culture on partitioned agar plates containing 5% sheep blood agar and McConkey agar number 3. The culture was positive onto blood agar plate after incubation at 37 °C overnight, with no swarming or hemolysis present; (**b**) Gram stain revealed a microorganism with the shape of *Bacillus*, with no obvious endospores, but if present, they did not alter the shape of the bacterium. Gram stain was variable.

**Figure 2 pathogens-11-01203-f002:**
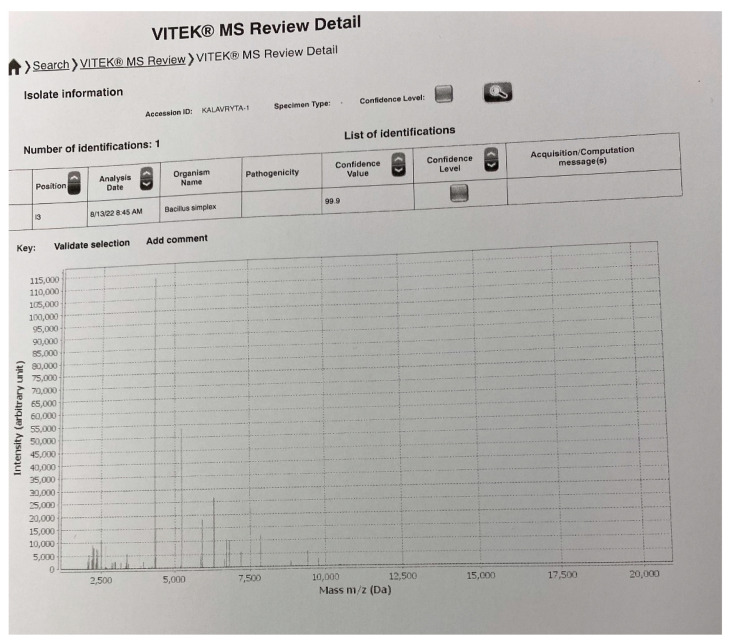
Spectral fingerprints obtained from the identification analysis of the culprit pathogen (MALDI-TOF Mass Spectrometry Vitek MS BioMerieux, Athens, Greece, BioMerieux Hellas). The strain was identified as *Bacillus simplex* with confidence value 99.9.

## Data Availability

Not applicable.

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
