# Peer review of "Bacillus simplex* as the Most Probable Culprit of Penetrating Trauma Infection: A Case Report"

_pathogens, 2022, doi:10.3390/pathogens11101203_

Round 1
Reviewer 1 Report (Previous Reviewer 2)
The manuscript is suitable for publication
Reviewer 2 Report (Previous Reviewer 1)
Accept in present form
This manuscript is a resubmission of an earlier submission. The following is a list of the peer review reports and author responses from that submission.
Round 1
Reviewer 1 Report
In the manuscript (pathogens-1924933) authors describe a clinical case that concerned an infection from Bacillus simplex bacterium. As, this is a rare case, the article is of interest. However, some technical issues must be addressed prior to manuscript publication. More details on the MALDI-TOF Mass Spectrometry method along with any spectral fingerprints obtained from the analysis must be given. The “Material and Methods” part should be reestablished. Instrumentation, magnification of samples in Figure 1b, etc can be added.
Other issues:
1. line 28: “B. simplex”. Please use italics.
2. line 55: “14300/mm3” change to “14300/mm3”.
3. line 130: “There are studies regarding B. cereus strains.” to “There are studies regarding B. cereus strains though.”
4. Is there a sequencing analysis available?
In general, I would like to recommend the publication of this manuscript in of Pathogens after the above Major issues are addressed.
Reviewer 2 Report
The manuscript by Panagiota Xaplanteri et al. describes an interesting case report on the identification of B. simplex involved in an orthopedic specimen.
The manuscript deserves numerous modifications and revisions that warrant rejection before possible resubmission.
The manuscript should use the passive turn of voice.
English typos should be corrected.
Treatment modalities with quinolone and clindamycin should be detailed.
Plates used for culture should be detailed (e.g., commercial manufacturer's contact information).
Bacterial names should be italicized.
MALDI TOF should be referenced as VitekMS ? Brucker ?
Reageant for disk diffusion susceptibility testing should be appropriately referenced.
Figure 1. explain the biplate nature of the agar.
Discussion: The entire discussion of virulence factors should be deleted because the case did not discuss them. Also, it might be more interesting to give details on the AST of Bacillus spp, since the authors detail the antibiotic treatment.